# Genome-Wide Identification and Characterization of MYB Gene Family and Analysis of Its Sex-Biased Expression Pattern in *Spinacia oleracea* L.

**DOI:** 10.3390/ijms25020795

**Published:** 2024-01-08

**Authors:** Zhilong Zhang, Zhiyuan Liu, Hao Wu, Zhaosheng Xu, Helong Zhang, Wei Qian, Wujun Gao, Hongbing She

**Affiliations:** 1State Key Laboratory of Vegetable Biobreeding, Institute of Vegetables and Flowers, Chinese Academy of Agricultural Sciences, Beijing 100081, Chinaqianwei@caas.cn (W.Q.); 2College of Life Sciences, Henan Normal University, Xinxiang 453007, China

**Keywords:** spinach MYB transcription factor, RNA-seq analysis, sex-biased expression

## Abstract

The members of the myeloblastosis (MYB) family of transcription factors (TFs) participate in a variety of biological regulatory processes in plants, such as circadian rhythm, metabolism, and flower development. However, the characterization of MYB genes across the genomes of spinach *Spinacia oleracea* L. has not been reported. Here, we identified 140 MYB genes in spinach and described their characteristics using bioinformatics approaches. Among the MYB genes, 54 were 1R-MYB, 80 were 2R-MYB, 5 were 3R-MYB, and 1 was 4R-MYB. Almost all MYB genes were located in the 0–30 Mb region of autosomes; however, the 20 MYB genes were enriched at both ends of the sex chromosome (chromosome 4). Based on phylogeny, conserved motifs, and the structure of genes, 2R-MYB exhibited higher conservation relative to 1R-MYB genes. Tandem duplication and collinearity of spinach MYB genes drive their evolution, enabling the functional diversification of spinach genes. Subcellular localization prediction indicated that spinach MYB genes were mainly located in the nucleus. Cis-acting element analysis confirmed that MYB genes were involved in various processes of spinach growth and development, such as circadian rhythm, cell differentiation, and reproduction through hormone synthesis. Furthermore, through the transcriptome data analysis of male and female flower organs at five different periods, ten candidate genes showed biased expression in spinach males, suggesting that these genes might be related to the development of spinach anthers. Collectively, this study provides useful information for further investigating the function of MYB TFs and novel insights into the regulation of sex determination in spinach.

## 1. Introduction

Myeloblastosis (MYB) transcription factors (TFs), one of the largest TF families in plants, bind to cis-acting elements in promoters through the MYB domain and further regulate the expression of their target genes, which play an important role in plant growth and development [1]. MYB TFs were first discovered in an avian leukosis virus, and analysis suggested that the conversion of c-myb to v-myb may be due to DNA rearrangement and the subsequent use of spliced RNA as an intermediate [2]. Abundant MYB genes have been found in animals, plants, and fungi, among others, and the structure and function of MYB genes have higher conservation in plants [3]. The first MYB-like TF discovered in plants was *ZmMYBC1*, which is involved in regulating anthocyanin synthesis in maize [4].

MYB TFs are named for their MYB structure, which usually has a DNA-binding domain, transcriptional activation domain, and negative regulatory region. The DNA domain located in the N-terminus of the MYB gene has a very high conservation type, and its basic unit is one to four incomplete repeated R structures (R1, R2, R3). The R structure is generally composed of 51–53 amino acid residues, each repeating forms of 3α helices, and the second and third helices form a regularly spaced helix–turn–helix (HTH) structure motif, in which HTH consists of three conserved tryptophan residues (W) and forms a hydrophobic center, with each tryptophan residue separated by 18 or 19 amino acids [5,6]. Because of the diversity of C-terminal sequences and the regulatory effect of the MYB conserved structure, MYB TFs have multiple functions.

According to the types and numbers of MYB TFs, they can be divided into four categories: 1R-MYB (MYB-related), 2R-MYB (R2R3-MYB), 3R-MYB (R1R2R3-MYB), and 4R-MYB [7]. The 4R-MYB TFs have four conserved domains that resemble R1/R2 structures, and this type of MYB is less present or even absent in plants. Currently, 4R-MYB proteins have been found in *Arabidopsis thaliana* (L.) Heynh. [8], *Vitis vinifera* L. [9], *Populus trichocarpa* Torr. & Gray. [10], and *Linum usitatissimum* Linn. [11], and their functions may be related to gametophyte and zygote development [12]. The 3R-MYB TFs have three R repeats, are generally more abundant in plants than the 4R-MYB, and are involved in the regulation of the cell cycle and cytokinesis in plants [13,14]. The 1R-RMYB TFs, containing only one or part of the MYB conserved domain, are the second largest group in the plant MYB TF family after the R2R3MYB subfamily. It can be divided into CCA1-like, CPC-like, TRF-like, TBP-like, I-box-binding-like, and R-R-type in *A. thaliana* [15]. The 1R-MYB TFs have received little attention, and their functions are mainly involved in maintaining the circadian rhythm [16], regulating cell morphogenesis [17], and participating in plant stress response to salt and cold [18]. The 2R-MYB TFs contain two MYB domains, R2 and R3, and they are the largest group in the MYB TF family of plants. They are widely distributed and have been reported in various plants, such as *A. thaliana* [8], *Oryza sativa* subsp. *japonica* Kato. [15], *Apocynum venetum* L. [19], *Daucus carota* var. *sativa* Hoffm. [20], and *Pennisetum glaucum* L. [21]. Their functions are complex and diverse, participating in various physiological and biological processes, such as primary and secondary metabolism [22,23,24], growth and development [25,26], the reproductive process [27,28], and biological and abiotic stress [29,30,31,32].

In addition to the MYB transcription factor family, there are other transcription factor families in plants, like the bHLH and WRKY transcription factor family. Studies have found that *IbbHLH33* has the function of improving plant cold resistance [33]. Moreover, bHLH TFs are also involved in anthocyanin synthesis in plants, flower development, and response to abiotic stresses, such as drought, salt, and cold [34]. It has been reported that the *HvWRKY1* gene plays an important role in the resistance of highland barley to leaf streak disease [35]. It can be concluded that transcription factors play a crucial role in plant growth and development, metabolic processes, disease resistance, and stress resistance. Therefore, it is of considerable importance to study transcription factor functions in plants.

As the reproductive organs of flowering plants, flowers produce the next generation through pollination, fertilization, and other processes. They are mainly composed of a calyx, petals, stamens, and pistils. The anther is an important part of the stamen, containing pollen; its development process is very delicate and complex and is affected by a variety of TFs and the external environment. Previous studies have shown that MYB TF is an extremely important TF in anther development, such as regulating the development of the tapetum layer [36], participating in pollen development [37], influencing the elongation of filaments, and participating in anther cracking [38,39]. In addition, MYB TFs can combine with other upstream and downstream TFs to regulate anther development through hormones, such as jasmonic acid (JA) [40], auxin [41], and gibberellin (GA) [42].

Spinach *Spinacia oleracea* L. is mostly dioecious with separate male and female individuals, although occasional monoecy with both male and female flowers has been observed. Thus, spinach is an ideal plant for studying the sex-determination mechanism [43,44]. MYB TFs play an important role in floral organ development; however, they remain poorly known in spinach. Here, we identified 140 MYB TFs and deciphered their structure, evolutionary relationships, and protein motif. We also found ten genes that have biased expression between female and male flowers, suggesting that these genes might be involved in flower organs. These results provide a broader understanding of the sex-determination mechanism in spinach.

## 2. Results

### 2.1. Identification of the MYB Gene Family in Spinach

A total of 140 MYB genes were identified using HMMER search in the spinach Sp_YY_v1 genome. Two genes were removed, as they did not contain R repeats. According to the number of R repeats, these MYB genes were divided into four categories: 4R-MYB (1), 3R-MYB (5), 2R-MYB (80), and 1R-MYB (54) (Appendix A).

Among the 140 MYB genes, there were 22 on chromosome 1, 26 on chromosome 2, 27 on chromosome 3, 20 on chromosome 4, 18 on chromosome 5, 21 on chromosome 6, and 6 on the 6 unassembled contigs. The number of MYB genes on each chromosome was similar and did not depend on the length of the chromosome. As shown in Figure 1, chromosome 5, with the shortest overall length, had 18 MYB genes, while chromosome 4, with the longest length, had only 20 MYB genes. The distribution of MYB genes on chromosomes 1, 2, 3, 5, and 6 was not well proportioned, and it was mainly concentrated in the 0–30 Mb region, with little distribution in the middle or end. However, the distribution of MYB genes on chromosome 4 showed a symmetrical form, and they were mainly distributed on both ends of chromosome 4 (Figure 1).

### 2.2. Phylogenetic Analysis and Classification of the Spinach MYB Gene Family

To infer the evolutionary relationship and corresponding functions of the 1R-MYB genes in spinach, we constructed a phylogenetic tree of 66 1R-MYB from *A. thaliana* and 54 1R-MYB from spinach (Figure 2). The 1R-MYB members were divided into 14 subgroups, in which the least subgroup contained only one member (S8, S12), and the most contained 20 members (S14). Not all subgroups contained 1R-MYB members of spinach and *A. thaliana*. S6, S8, S11, and S12 contained only the 1R members of spinach. However, S10 contained only the 1R-MYB members of *A. thaliana*. In addition to these five subgroups, which contained members of only one species, the other seven subgroups all contained members of the 1R-MYB of the two species. Within these seven subgroups, members of the two species were not evenly distributed. Although the S2 and S7 subgroups had the same total number of members, they did not contain the same amount of spinach or *A. thaliana*. In addition, the other subgroups not only had different proportions of spinach and *A. thaliana* but also a different total number of each subgroup.

We constructed a phylogenetic tree using protein sequences from 133 R2R3-MYB genes of *A. thaliana* and spinach 2R (80), 3R (5), and 4R (1) MYB genes (Figure 3). A total of 219 genes were divided into 26 subgroups according to the clustering branches of the phylogenetic tree, among which subgroups S1, S2, S3, and S4 contained only MYB genes in spinach. In subgroup S1, there were four 2R-MYB genes and one 3R-MYB gene, and only 4R-MYB in spinach was also in this subgroup. This suggests that these genes may play a unique role in spinach. Meanwhile, S5 contained only the ATMYB_91 of *A. thaliana*, suggesting that the role of this gene in *A. thaliana* may not have gene expression with similar functions in spinach. The remaining 3R-MYB genes in spinach were located in the S6, S7, S11, and S17 subgroups. In addition, the other four genes in the S6 subgroup, except one spinach 3R-MYB gene, were *A. thaliana* genes. The other 2R-MYB genes in spinach were clustered into different subgroups, with different genes in *A. thaliana*.

### 2.3. Analysis of the Conserved Motifs, Domains, and Structure of Spinach MYB Genes

By analyzing the conserved motifs, conserved domains, and structures of MYB genes in spinach combined with the grouping of phylogenetic trees, the evolutionary relationships among MYB genes in spinach were analyzed (Figure 4). Interestingly, all 1R-MYB genes, except *YY_121510*.*1*, contained motif 2, indicating that this motif was extremely conserved in 1R-MYB, followed by motif 9, which was present in most members, and motif 1 was also relatively conserved. The other motifs were distributed in several genes. Of the 54 1R-MYB genes, 37 contained the SANT superfamily, and some even contained two of the superfamilies, indicating that this conserved domain was extremely conserved and important in the 1R-MYB gene. In the analysis of the gene structure of these sequences, we found that the structure of each of the spinach 1R-MYB genes varied greatly, with the number of CDSs ranging from 1 to 14, among which 6 genes had only 1 or no intron, indicating that these genes may have been transcribed very frequently. The structure of genes was somewhat similar to that of the MYB genes that clustered in the same subgroups. The diversity of the 1R-MYB protein sequence in spinach was also found in the 1R repeat sequence alignment (Appendix A).

In these analyses, 4R-MYB contained two conserved motifs, 3 and 6; two conserved domains; and 16 CDSs. There were five genes in 3R-MYB: one gene contained four conserved domains, one contained three conserved domains, and the other three contained two conserved domains. The conserved domains were very similar. Genes containing four domains had the most CDSs (22), with one having 15 CDSs and the other three each having 11 CDSs. In terms of total length, the average length of the 3R-MYB genes was the longest among the MYB genes in spinach.

From the results of the analysis, 80 genes in 2R-MYB had relatively identical motif species and relative positions, with motifs 1–6 having high conservation among these genes. In all 2-RMYB genes, the conserved domains were mainly in the PLN03091 superfamily, followed by the SANT superfamily and the PLN03212 superfamily. The gene structure showed that the number of 2R-MYB CDSs ranged from 2 to 19, but genes containing 2 or 3 CDSs were in the majority, including 43 genes containing 3 CDSs and 15 genes containing 2 CDSs (Figure 5). Multiple sequence alignment showed that the R2R3-MYB gene had a higher conserved R repeat sequence than the others (Appendix A).

### 2.4. Subcellular Localization and Cis-Acting Element Analysis of Spinach MYB Genes

To understand the function of the MYB gene family in spinach, we performed subcellular localization predictions for all MYB gene expression sites. Of the 140 MYB genes, 126 (90%) were expressed in the nucleus. Seven were expressed in the cytoplasm, namely *YY_075510*.*1*, *YY_076070*.*1*, *YY_084880*.*1*, *YY_116950*.*1*, *YY_120510*.*1*, *YY_221390*.*1*, and *YY_236230*.*1*, accounting for 5% of all MYB genes. There were four possible expressions in chloroplasts: *YY_002720*.*1*, *YY_163440*.*1*, *YY_241060*.*1*, and *YY_245610*.*1*. *YY_040140*.*1* and *YY_248600*.*1* were expressed in the mitochondria, and *YY_020210*.*1* was expressed in the plasma membrane (Appendix A; Appendix A).

The prediction results of cis-acting elements showed that the promoter elements of the MYB gene in spinach mainly included hormone response elements (abscisic acid, salicylic acid, gibberellin, methyl jasmonate, auxin, and flavonoids), light response elements, defense and stress response elements, and components involved in cell cycle regulation and plant meristem regulation. Among these elements, photoresponsive elements were the most numerous, followed by hormones, of which abscisic acid, gibberellin, and salicylic acid-related elements were the most common (Figure 6).

In summary, the vast majority of MYB transcription factors in spinach should regulate the expression of corresponding genes by binding with cis-acting elements in the nucleus to play a regulatory role in certain traits or biological processes. The different expression locations of transcription factors may be greatly related to the function of regulated gene generation.

### 2.5. Tandem Duplication and Collinearity Analysis of Spinach MYB Genes

Six tandem duplicate gene pairs were found in the MYB genes of spinach, among which *YY_002590*.*1* and *YY_002600*.*1*, *YY_075500*.*1* and *YY_075510*.*1*, and *YY_066510*.*1* and *YY_066520*.*1* were tandem gene pairs. However, four genes, *YY_037380*.*1*, *YY_037390*.*1*, *YY_037400*.*1*, and *YY_037410*.*1*, were linked together (Figure 1). In addition, intra-species collinear analysis showed that there were 12 collinear gene pairs related to the MYB gene, of which seven pairs belonged to the MYB gene family, and the remaining five pairs corresponded to one gene locus on chromosome 1 and ctg000064_np121212 and three gene loci on chromosome 6 (Figure 7). Of the seven MYB collinear gene pairs, four were located entirely on the assembled chromosomes: *YY_009230*.*1* and *YY_037380*.*1*, *YY_003840*.*1* and *YY_180090*.*1*, *YY_010200*.*1* and *YY_208670*.*1*, and *YY_105030*.*1* and *YY_121280*.*1*. Among the other three pairs, *YY_244550*.*1* and *YY_244770*.*1* were located on different contigs; *YY_127100*.*1* and *YY_244160*.*1* were located on one chromosome; and *YY_180870*.*1*, and *YY_245450*.*1* were located on one contig. The collinearity of genes in the genome indicates that collinear genes had highly similar DNA sequences. However, the MYB gene was collinear with other non-MYB genes, indicating that, although their DNA sequences were similar, their protein sequences showed diversity, which may be due to changes in the gene-coding region, resulting in changes in the protein sequences of genes and different functions.

To elucidate the evolutionary relationship between the MYB gene family of spinach and other species, we performed a collinearity analysis of the spinach genome with *A. thaliana*, sugar beet, and rice (Figure 8). There were 65 MYB genes in spinach and 95 collinear gene pairs in *A. thaliana*. Among these, *YY_032040*.*1* had four collinear gene pairs in *A. thaliana*, and *YY_160940*.*1* and five other MYB genes had three collinear genes in *A. thaliana* and 17 genes with two collinear gene pairs. The remaining 42 were separate collinear gene pairs. In the collinearity analysis with sugar beet, there were 79 MYB genes in spinach and 89 collinear gene pairs in sugar beet, of which 10 MYB genes had two collinear gene pairs each. In rice, 35 gene pairs were collinear with the MYB gene family of spinach, and 30 MYB genes were collinear. Among these, three genes were collinear with *YY_098920*.*1*, and two gene pairs were collinear with *YY_095350*.*1*, *YY_097110*.*1*, and *YY_127100*.*1*. The remaining 26 genes each had one gene in common. According to the differences in the number of MYB genes involved in *A. thaliana*, sugar beet, and rice, the relatives of MYB genes in spinach and the three species were sugar beet, *A. thaliana*, and rice in order from strong to weak. Surprisingly, we analyzed the MYB genes of spinach involved in the three species, and there were 19 MYB genes involved in the three species together, which indicates that these 19 spinach MYB genes were extremely conserved and that they may play the same function in the four species (Figure 9).

### 2.6. Sex-Biased Expression Pattern of MYB Genes

According to the morphology and degree of flower organ development, previous studies [45] have divided the development of spinach flowers into five main stages (stages 1–5). In the first stage, the size of the flower buds ranges from 0.2 to 0.5 mm, and the pistil and stamen form two opposite sepal primordia on the periphery of the meristems. When the flower buds develop to 0.5–1 mm, it is called the second stage. The sepals increase with it, and the distal end of the sepals of the pistil surrounds the flower meristem. In the third stage, the sepal primordia grow, four stamen primordia form in the periphery of the central dome, a female flower sepal covers the central dome, and the dome begins to differentiate into an ovary. In the fourth stage, the stamen primordium generally develops into four anthers, the central region of the ovary forms a pistil, and the internal ovule differentiates in the ovary. Finally, the anther and ovule mature and the stigma protrudes from the sepal closure, all of which comprise the fifth stage. In general, a gene was highly expressed in a certain tissue or at a certain time in the tissue, indicating that the gene was involved in the development of the relevant tissue at the corresponding time. By comparing the expression levels of MYB genes in different stages of male and female flowers of spinach, it is expected to find MYB genes related to sex development in spinach.

The results showed that 105 differentially expressed genes (DEGs) were found through comparison (Figure 10). According to the expression heat map, 33 genes had male-biased expression and may be involved in anther development. For example, eight genes, such as *YY_141510*.*1* and *YY_183170*.*1*, were highly expressed in the MS5 stage. Six genes, such as *YY_155030*.*1* and *YY_208330*.*1*, were highly expressed in MS4. Fifteen genes, such as *YY_175740*.*1* and *YY_216230*.*1*, were highly expressed in MS3, with the largest numbers. Of these, three genes were highly expressed in MS3 and continued to be expressed in MS4, and two were slightly less expressed during early anther development (MS1 and MS2) and then highly expressed in MS3. Finally, three genes, including *YY_208670*.*1*, were highly expressed in MS1.

Among the 33 genes mentioned above, we identified 10 candidate genes (*YY_141510*.*1*, *YY_155030*.*1*, *YY_123170*.*1*, *YY_183170*.*1*, *YY_055300*.*1*, *YY_016090*.*1*, *YY_175740*.*1*, *YY_216230*.*1*, *YY_208330*.*1*, and *YY_121280*.*1*) that might determine anther development, as their homologous function has been confirmed in *A. thaliana* (Appendix A). Three genes, *YY_141510*.*1*, *YY_155030*.*1*, and *YY_123170*.*1*, were clustered with the S18 subgroup members of *A. thaliana* in phylogenetic tree analysis, suggesting that they may have the same function. Studies have shown that ATMYB_65 and ATMYB_33 of this subgroup are related to programmed cell death in the tapetum layer during anther development in *A. thaliana* [46]. *YY_183170*.*1* is homologous to ATMYB_125 (DUO1) in *A. thaliana*, and relevant studies have shown that this gene is involved in spermatocyte differentiation and pollen development [37]. In addition, homologous gene ATMYB_125 (DUO1) in rice is involved in pollen development [47]; therefore, we believe that this gene may have the same function in spinach. *YY_055300*.*1* belongs to the S19 subgroup of *A. thaliana*, two members of which, ATMYB_21 and ATMYB_24, are related to stamen development and are regulated by GA and JA [48]. *YY_016090*.*1* is homologous to ATMYB_108, regulated by GA and JA in *A. thaliana*, and correlated with stamen development and anther maturation [49]. *YY_175740*.*1* and *YY_216230*.*1* may be ATMYB_35 (TDF1) homologs involved in the regulation of the tapetal layer in *A. thaliana* [50,51]. *YY_208330*.*1* may have the same function as the S6 gene in *A. thaliana* and may participate in the phenylpropane pathway and flavonol synthesis [52]. Studies have shown that the phenylpropane pathway can be involved in anther development [53]. *YY_121280*.*1* is a homolog of the TF ATMYB_103, and termination of a single base in the R2R3 region of ATMYB_103 leads to male sterility in *A. thaliana* [54]. Thus, loss of function of this gene may lead to male sterility in spinach.

## 3. Discussion

Spinach (*S*. *oleracea* L.) is an annual wind-pollinated plant and a member of the family Amaranthaceae in the order Caryophyllales [55]. With the deepening of the understanding of the importance of spinach, an increasing number of researchers are paying attention to its edible and scientific research value. The analysis of MYB gene family members in spinach is a fast and effective way to quickly understand the effects of the MYB gene family on spinach. Combined with the transcriptome data of flower organs in different periods, MYB genes related to anther development can be quickly screened out. Further study of these MYB genes is beneficial to the genetic breeding of spinach, to enrich the germplasm resources of spinach, and to lay a foundation for the investigation of anther development in spinach.

First, using the hidden Markov model of the MYB gene family, 140 members of the spinach MYB gene family containing R repeats were identified using a HEMMER search, of which 54 were 1R-MYB, 80 were 2R-MYB, 5 were 3R-MYB, and 1 was 4R-MYB. We speculated that the chromosome architecture may account for the different distributions of MYB genes between autosomes and sex chromosomes because we previously found one end of each autosome and the middle region of the sex chromosome pair to be rich in repeats [56]. This phenomenon can also be seen as a means for plants to protect themselves. The role of sex chromosomes in sexual reproduction in plants is self-evident; however, the evolution of male sex chromosomes (Y) in plants is accompanied by the continuous accumulation of recombination inhibition, which leads to the continuous weakening of the role of Y chromosomes in plants [57]. To better adapt to the environment, this process is relatively slow. Having genes distributed on both ends of the sex chromosome also helps to reduce the degree of phenotypic differences between individuals due to gene loss.

Since there was only one or part of an R repeat in the 1R-MYB gene family and because there are few related studies [58], we separated spinach 1R from other (2R, 3R, 4R) members and *A. thaliana* MYB genes to construct a phylogenetic tree using the NJ method to better understand the evolutionary relationship between them. However, in both 1R and 2R, only MYB genes in spinach were grouped, suggesting that these genes are unique in spinach. In the 2R phylogenetic tree, the 3R members did not cluster, suggesting that the origin of the 3R members was not from the 3R genes but may have evolved from the 2R members, consistent with a previous study [5]. Structurally, 2R-MYB has two R repeats, and the R2 and R3 MYB domains are highly conserved [59], which is one of the reasons why 2R-MYB genes are more conserved. From an evolutionary point of view, studies have shown that 2R-MYB transcription factors are distributed in fungi, animals, and plants, indicating that 2R-MYB transcription factors are relatively ancient and conserved [60]. The abundance of 2R-MYB genes is predominant in both *A. thaliana* and spinach, and this discovery has also been found in *Rhododendron delavayi* [61], mainly because the 2R-MYB transcription factor is amplified in higher plants, increasing the number of 2R-MYB genes. In combination with an evolutionary perspective, a relatively conservative structure that has a relatively conserved function may be related to the stability of the plant’s genetic material and its ability to adapt to the environment with more favorable traits [62].

In the analysis of the conserved motif, conserved domain, and structure of these genes, combined with multiple sequence alignment of R repeats, the differences among 1R members were large, and the gene functions were diversified. Compared with 1R, 2R members were more conservative, in which R2 repeats were the most conservative, and the first tryptophan (W) in R3 repeats was mostly replaced by phenylalanine (F) and isoleucine (I), showing diversity. Notably, we found that the SANT superfamily and PLN03212 superfamily were conserved domains in 1R-MYB, 2R-MYB, and 3R-MYB. It included most of the 1R members of the SANT superfamily conservative domain, such as *YY_021170*.*1* and *YY_138730*.*1* in 2R-MYB and *YY_055340*.*1* and *YY_088780*.*1* in 3R-MYB. *YY_055660*.*1* and *YY_175020*.*1* in 1R-MYB, *YY_055300*.*1* and *YY_177300*.*1* in 2R-MYB, and *YY_088780*.*1* in 3R-MYB had the PLN03212 superfamily domain. These two domains were highly conserved in the spinach MYB gene family, playing a very important role.

Tandem duplication and the collinearity of genes are considered one of the driving forces of species evolution [63]. Similarly, there are many tandem repeat gene pairs in the evolution of the spinach genome, including the spinach MYB gene. Collinearity analysis of the whole genome of spinach showed that the MYB gene in spinach had a collinearity relationship not only with members of the MYB gene family but also with other genes, indicating that the MYB genes may change their structure and have other functions in other ways, such as base mutation and loss. We analyzed the collinearity between spinach MYB genes and *A. thaliana*, sugar beet, and rice. There were 65 spinach MYB genes involved in collinearity with *A. thaliana*, 79 with sugar beet, and 30 with rice. This suggests that spinach was most closely related to sugar beets, followed by *A. thaliana* and rice. This is because spinach and sugar beets belong to the family Amaranthaceae, while *A. thaliana* belongs to the family Brassicaceae. Spinach, sugar beet, and *A. thaliana* are dicotyledonous plants, while rice is monocotyledonous. The 19 MYB genes found in all four species indicate that these genes are very conserved and may have the same biological function in all four species, playing a unique role in the plant. Relevant studies have shown that short tandem repeats in the genome can regulate the expression level of related genes [64]. Tandem repeats have the same sequence at the beginning, which manifests as an increase in copy number in the genome. However, with the continuous occurrence of gene replication, the bases in some regions will change. This leads to a change in the function of the gene [65]. However, tandem duplication may also be related to the emergence of specific biological functions in plants; for example, the first specialized gene in the caffeine and saffron synthesis pathways in coffee and gardenia, *GjCCD4a*, recently evolved through tandem gene replication in two different genera [66]. Moreover, tandem duplication, in addition to its role in metabolism, is involved in pathogenic bacterial reactions [67] and the formation of specific plant functions (such as enzymes related to meat feeding in pitcher plants) [68]. Therefore, the tandem duplication of the MYB gene in spinach may also have some special functions.

To screen sex-biased MYB genes in spinach, we analyzed MYB gene expression levels in female and male flowers of spinach during different periods. A total of 33 differentially expressed genes were screened, comprising 24 2R-MYB genes, 8 1R-MYB genes, and 1 3R-MYB genes, which were only expressed in stamens and not in pistils. The 1R-MYB gene in spinach is involved in reproduction, which is consistent with the results of the function of the MYB gene in *A. thaliana* [5]. With further analysis of the sex chromosomes of spinach, there was a 24.1 Mb Y-linked region on the Y chromosome, of which 10 Mb was Y specific [56]. To more comprehensively search for MYB genes related to the sex of spinach, we found two MYB genes in the sex-linked region of spinach: *YY_140780*.*1*, located in the IV2-1 region, and *YY_141510*.*1*, located in the IV2-2 region. *YY_140780*.*1* belongs to 1R MYBs, and its homologous gene in *A. thaliana* is *AT2G31970*, RAD50; it is involved in double-strand break repair. It is a component of the meiotic recombination complex that processes meiotic double-strand breaks to produce single-stranded DNA ends, which act in the homology search and recombination and accumulate in the nucleus during meiotic prophase, a process regulated by PHS1. In addition, the mutation of this gene will cause plant sterility and produce empty hornfruit. However, it is not specifically involved in meiosis but exists in tissues that divide rapidly, such as young root tips and stem tips [69]. Since meiosis occurs early in both pistils and stamens in spinach, the expression of this gene is high and decreases with the end of division, which is consistent with the results of previous studies. *YY_141510*.*1* is a member of 2R-MYB, and its homologous gene is ATMYB_101. Data from the Plant Public RNA-seq Database (https://plantrnadb.com/ (accessed on 16 September 2023)) showed that ATMYB_101 was mainly expressed in pollen and during the late flowering period, in line with our expectations. Similar to the other eight candidate genes, it may participate in stamen development in spinach by affecting the growth state and corresponding function of certain anther tissues. In conclusion, we suggest that these ten MYB genes may be involved in the formation of different sexes in spinach.

## 4. Methods

### 4.1. Identification of the MYB Gene Family in Spinach

To identify MYB family members in *Spinacia oleracea* L., the MYB domain HMM profile (PF00249) of the myb DNA-binding domain from the Pfam database (http://pfam.sanger.ac.uk/ (accessed on 23 May 2023)) was used as a query to perform an HMMER search within the spinach genome with an E-value cut-off of 1 × 10^−5^ following the HMMER User Guide. For further identification, we searched for the R number of all sequences to verify whether they belonged to the MYB gene family and classified them by R number.

### 4.2. Phylogenetic Analysis and Classification of the Spinach MYB Gene Family

To better classify the spinach MYB gene family, we analyzed it in two parts; the first part consisted of 1R members, and the second part consisted of all members except 1R members, including 2R, 3R, and 4R. The two-part analysis method was the same as shown in the second part, and we downloaded the *A. thaliana* R2R3-MYB gene family protein sequences from the Plant Transcription Factor Database (http://planttfdb.gao-lab.org/ (accessed on 23 May 2023)). Full-length sequences of MYB gene family members that included more than two repeats and the R2R3-MYB proteins of *A. thaliana* were used for multiple sequence alignment using the MUSCLE method of MEGA 7 (https://www.megasoftware.net/show_eua (accessed on 24 May 2023)) and were modified through Multiple Alignment Trimming in TBtools (https://github.com/CJ-Chen/TBtools-II/releases (accessed on 27 April 2023)) [70], followed by evolutionary tree construction using the neighbor-joining method. Finally, the phylogenetic tree of spinach MYB genes was generated using TVBOT in Chiplot (https://www.chiplot.online/tvbot.html (accessed on 3 June 2023)) [71].

### 4.3. Analysis of the Conserved Motifs, Domains, and Gene Structure of the Spinach MYB Genes

To understand the structure of spinach MYB genes, the online program MEME (https://meme-suite.org/meme/tools/meme (accessed on 24 May 2023)) was applied to analyze the conserved motifs [72]. The resulting sequences were subjected to the NCBI Conserved Domain Search (CCD, https://www.ncbi.nlm.nih.gov/cdd (accessed on 24 May 2023)), and the MYB domain was confirmed in all identified proteins. The gene structure, including the exon, intron, and untranslated region (UTR) of the genes, was displayed in TBtools according to the gff3 file of the genome. Based on the identification of the MYB transcription factors, the motifs and domains of the MYB genes were visualized using TBtools software [73].

### 4.4. Subcellular Localization and Cis-Acting Element Analysis of the Spinach MYB Genes

The subcellular localization of the spinach MYB gene was predicted using a subcellular localization prediction site (https://wolfpsort.hgc.jp/ (accessed on 5 June 2023)). The online website Evenn (http://www.ehbio.com/test/venn/#/ (accessed on 6 June 2023)) was used to visualize the predictions [74]. The online website PlantCARE (http://bioinformatics.psb.ugent.be/webtools/plantcare/html/ (accessed on 10 August 2023)) was applied to predict the cis-acting elements within the 2000 bp upstream of all spinach MYB genes [75]. The position, type, and number of cis-acting elements in the sequence were visualized using TBtools software.

### 4.5. Location of the Spinach MYB Genes on the Chromosome, Tandem Duplication, and Collinearity Analysis

The length, start positions, and end positions of the spinach MYB genes were obtained from the genome and GFF3 files, which were visualized using Gene Location Visualize from GTF/GFF in TBtools. We used TBtools analysis of spinach MYB gene collinearity with Circos to visualize segmental-duplicated gene pairs [76]. At the same time, the *A. thaliana*, *Oryza sativa* L., and *Beta vulgaris* L. genomes and gff3 files were downloaded from the Ensembl Plants website (http://plants.ensembl.org/index.html (accessed on 6 June 2023)). The syntenic relationships between the spinach MYB genes and genes from *A. thaliana, O*. *sativa*, and *B. vulgaris* were determined using TBtools [77].

### 4.6. RNA-seq Analysis

The RNA-seq data used in this study were generated in a previous study [45]. RNA was extracted from the flower organs of dioecious spinach at different stages. Transcriptome data were deposited in the NCBI database under BioProject number PRJNA724923 (https://www.ncbi.nlm.nih.gov/bioproject/?term=PRJNA724923 (accessed on 24 August 2023)). We first filtered the raw RNA-seq reads using fastp (v0.20.0) [78] and then aligned the clean reads to spinach reference Sp_YY_v1 [56] using HISAT2 (v4.8.2) [79] with default parameters. Read counts per gene were estimated using the feature Counts (v2.0.1) [80] and converted to transcripts per million (TPM). DEGs were identified between female and male flowers using the R package DESeq (v1.14) [81] with *p* < 0.05 and fold change (FC) > 2.

## 5. Conclusions

In this study, we first identified the members of the spinach MYB gene family and then constructed a phylogenetic tree using the NJ method combined with *A. thaliana* MYB genes to analyze the evolutionary relationship of spinach MYB genes and the conserved motifs, conserved domains, and gene structures contained in each sequence. We found that the spinach MYB genes 1R-MYB, 2R-MYB, and 3R-MYB all contained SANT superfamily and PLN03212 superfamily conserved domains. The R2R3-MYB gene family had the highest conservation in spinach. Spinach MYB genes were mainly in the nucleus and involved in the synthesis of auxin, gibberellin, abscisic acid, circadian rhythm, cell differentiation, reproduction, and flower development. Tandem repetition and collinearity diversified the functions of spinach genes and promoted evolution in the spinach genome. Collinearity analysis with *Arabidopsis*, sugar beet, and rice showed that 19 genes were collinear in all three species, indicating that these 19 genes had high conservation. Finally, we combined the transcriptome data of male and female organs of spinach at different periods to screen ten candidate genes that may be related to anther development, which laid a foundation for the functional analysis of spinach MYB gene in spinach anthers. However, the analysis of the MYB transcription factor family in spinach further confirmed the complex and diverse biological functions of this transcription factor family in plants. At the same time, this study also provides relevant information for genetic breeding and the molecular basis of sex differentiation in spinach plants.

## Figures and Tables

**Figure 1 ijms-25-00795-f001:**
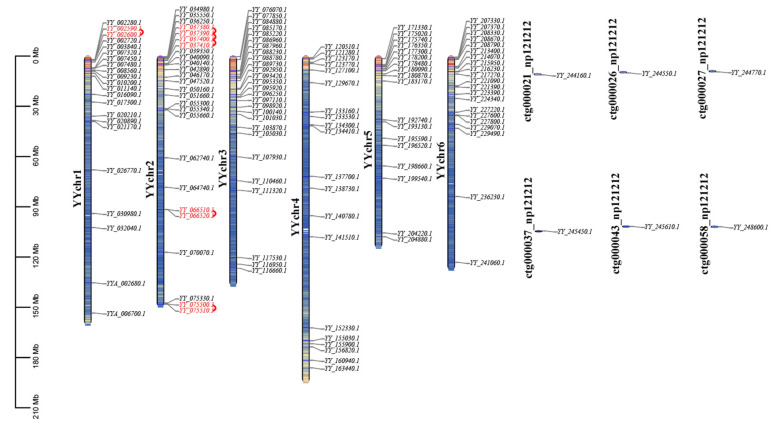
Distribution of 140 MYB genes among six chromosomes in the Sp_YY_v1 assembly. A total of 140 genes were mapped to six chromosomes. The tandem duplicated pairs are indicated in red. Chromosomes are filled with gene density. The data were visualized using TBtools (v2.030).

**Figure 2 ijms-25-00795-f002:**
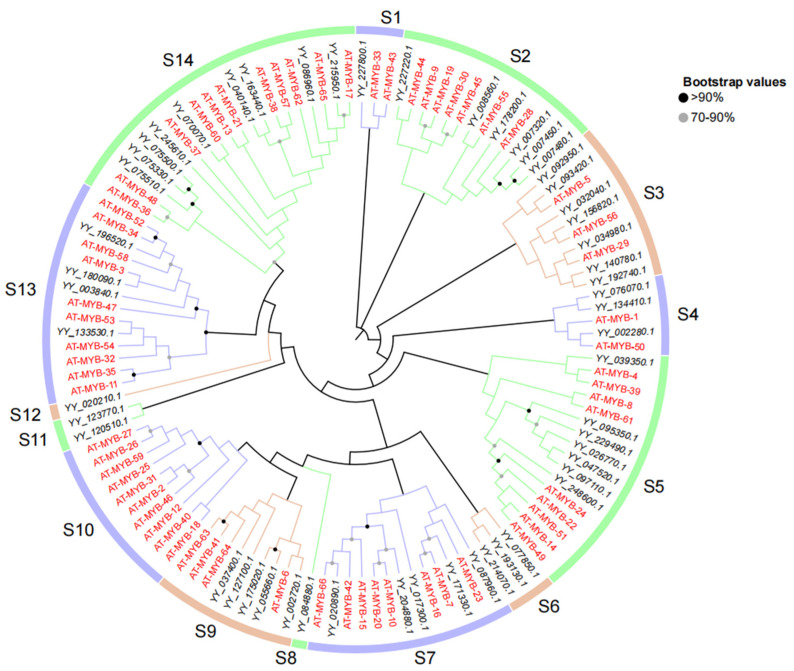
Phylogenetic tree of 1R-MYB from spinach and *A. thaliana*. The 1R-MYB (54) genes in spinach are marked in black, while those (66) in *A. thaliana* are labeled in red. The different colored lines in the outermost ring represent different classification groups. The corresponding *A. thaliana* MYB gene ID in the figure is shown in Appendix A. Protein multiple sequence alignment and visualization were performed using MEGA7 (v7.0.26) and modified through Multiple Alignment Trimming in TBtools, and data beautification was performed using TVBOT in Chiplot.

**Figure 3 ijms-25-00795-f003:**
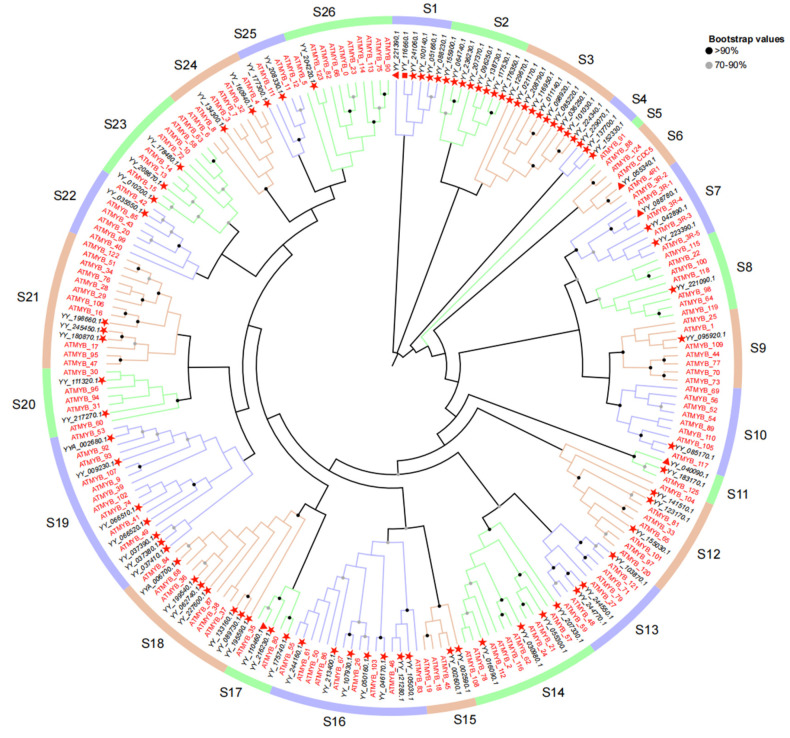
Phylogenetic tree of 2R-MYB, 3R-MYB, and 4R-MYB genes in spinach and 2R-MYB genes in *A. thaliana*. The genes in spinach are marked in black, while those in *A. thaliana* are marked in red. Genes with a red star belong to 2R-MYB. Those with a triangle belong to 3R-MYB, and the square represents 4R-MYB in spinach. The different colored lines in the outermost ring represent different classification groups. The corresponding *A. thaliana* MYB gene ID in the figure is presented in Appendix A. Protein multiple sequence alignment and visualization were performed using MEGA7 (v7.0.26) and modified through Multiple Alignment Trimming in TBtools, and data beautification was performed using TVBOT in Chiplot.

**Figure 4 ijms-25-00795-f004:**
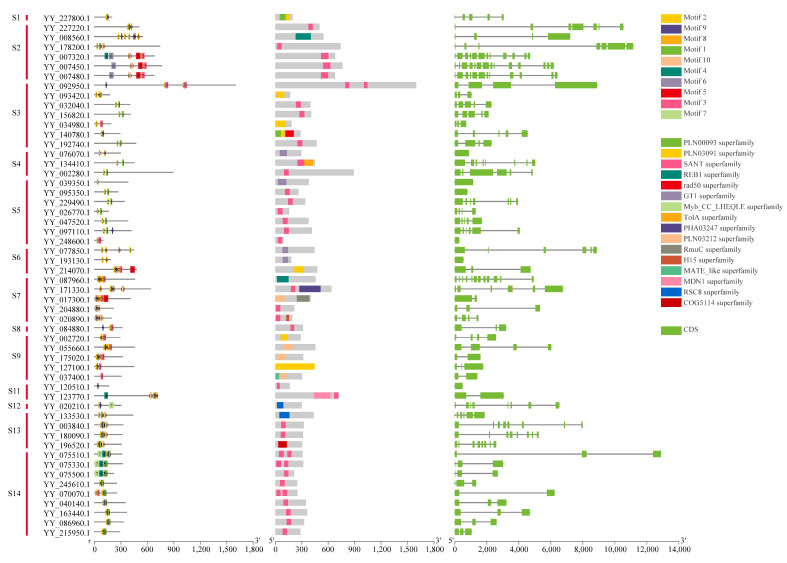
Conserved motif, conserved domain, and structure analysis of spinach 1R-MYB genes. Different color blocks represent different conservative motifs, and the corresponding color blocks have relative numbers. Different colors in the conserved domain also represent different domains. For the gene structure of 1R-MYB, the green box represents CDSs, and the horizontal line in the box represents introns. Motifs were found using the MEME online website, and conservative domains were found using the NCBI CD-search website. The software used for final visualization was TBtools.

**Figure 5 ijms-25-00795-f005:**
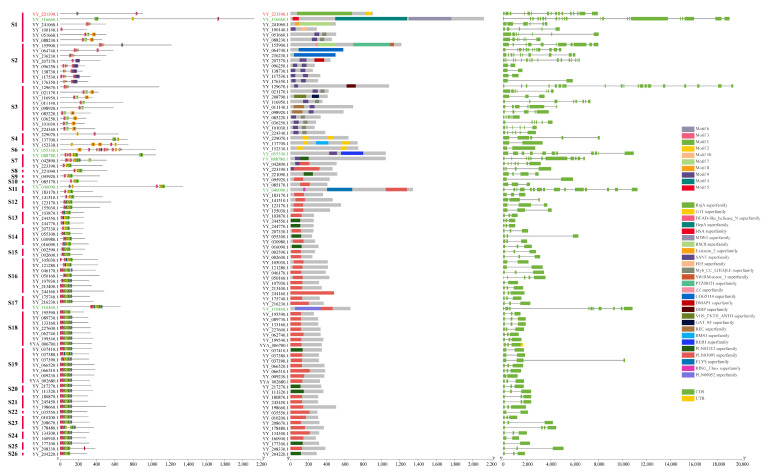
Conserved motif, conserved domain, and structure analysis of spinach 2R-MYB, 3R-MYB, and 4R-MYB genes. The gene ID in red represents 4R-MYB, green represents 3R-MYB, and black represents 2R-MYB. Different color blocks represent different conservative motifs, and the corresponding color blocks have relative numbers. Different colors in the conserved domain also represent different domains. In the gene structure, the green box represents CDSs, the yellow box represents UTRs, and the horizontal line in the box represents introns. Motifs were found using the MEME online website, and conservative domains were found using the NCBI CD-search website. The software used for final visualization was TBtools.

**Figure 6 ijms-25-00795-f006:**
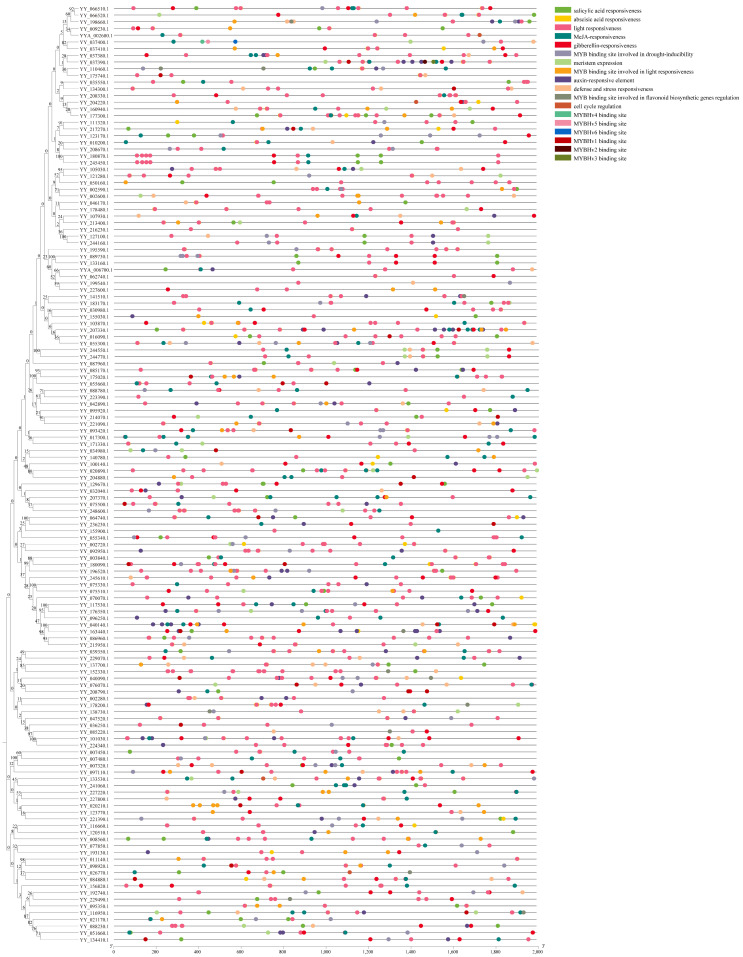
Cis-acting element analysis of spinach MYB genes. Different colors on the black lines represent different elements of the promoters. The online website PlantCARE was used to predict the cis-acting elements, which were then visualized using TBtools software.

**Figure 7 ijms-25-00795-f007:**
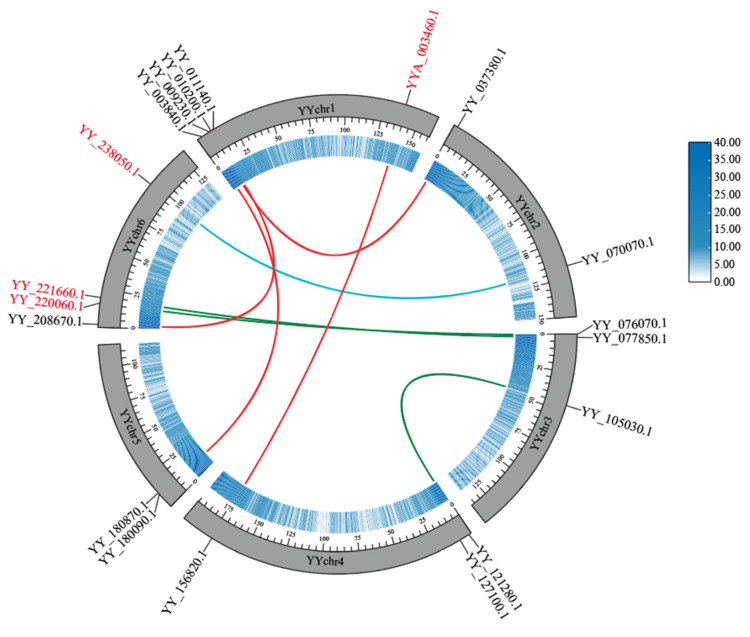
Intraspecific collinearity of the spinach genome. The black gene ID is the spinach MYB gene, while the red gene ID represents the other genes. Different colored lines represent collinear gene pairs on different chromosomes. The inner blue circle indicates the gene density of the corresponding chromosome. The data were visualized using TBtools.

**Figure 8 ijms-25-00795-f008:**
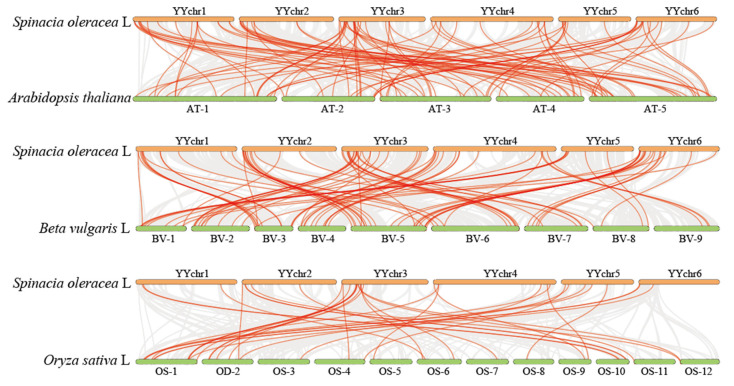
Genomic collinearity analysis of spinach and *A. thaliana*, sugar beet, and rice. The genomes of spinach and the three species are collinear. Collinear gene pairs are connected by gray lines, and red lines represent collinear gene pairs with the spinach MYB gene. BV: *Beta vulgaris* L, AT: *Arabidopsis thaliana*, OS: *Oryza sativa* L. The data were visualized using TBtools.

**Figure 9 ijms-25-00795-f009:**
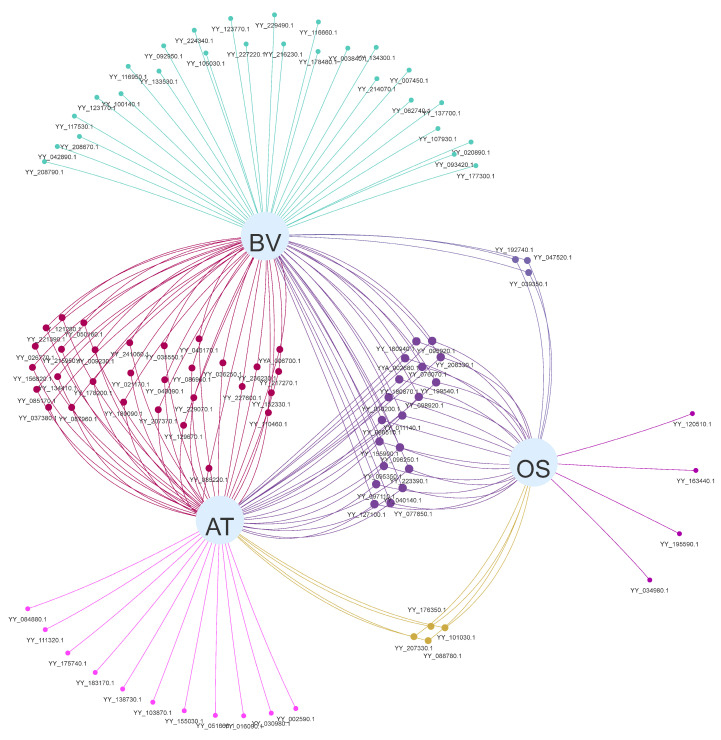
Relationships of spinach MYB genes involved in collinearity with each of the three species. Different points represent different genes involved in collinearity. The lines between the dots and the gray circles indicate the presence of collinear gene pairs, and the number of lines represents the collinear numbers involved in the corresponding species. A single connected line indicates that the gene is collinear only in that species, and that the same collinear gene pairs of two or more species have the same color. BV: *Beta vulgaris* L, AT: *Arabidopsis thaliana*, OS: *Oryza sativa* L. Collinear gene pairs were identified using TBtools and visualized via Evenn online.

**Figure 10 ijms-25-00795-f010:**
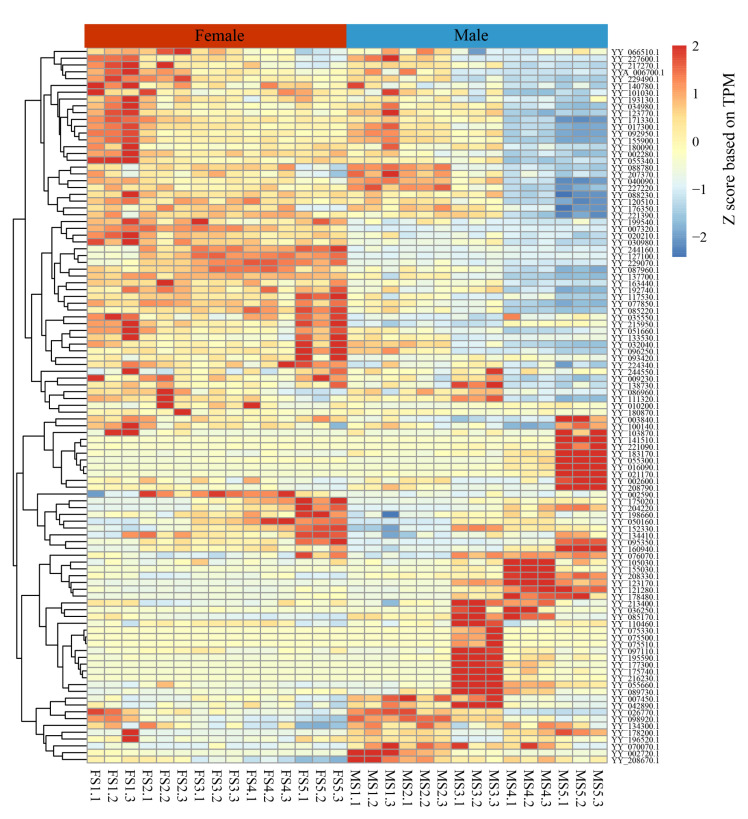
Expression patterns of the MYB gene in male and female flower organs of spinach in five different periods. FS1–5: 1–5 stages of pistil development. MS1–5: 1–5 stages of stamen development. FS1.1, FS1.2, and FS1.3 represent three repetitions in the same period. DEGs were identified between female and male flowers using the R package DESeq (v1.14) with *p* < 0.05 and fold change (FC) > 2.

## Data Availability

All data are contained within the article and Appendix A.

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
