# Peer review of "Genome-Wide Identification and Characterization of MYB Gene Family and Analysis of Its Sex-Biased Expression Pattern in Spinacia oleracea L."

_ijms, 2024, doi:10.3390/ijms25020795_

Round 1

Reviewer 1 Report

Comments and Suggestions for Authors

The manuscript is prepared on a current and interesting topic. I recommend the following edits before publishing it: Introduction - it would be good to add a general paragraph about the importance of transcription factors for plants and resistance to abiotic and biotic stress (e.g. bHLH - Yu et al. (2022) (10.17221/115/2021-CJGPB) - cold; Muhammad et al. (2023) (10.17221/2/2022-CJGPB) - overview of function; WRKY - Jing et al. (2023) (10.17221/14/2023-CJGPB) - diseases, etc.). Materials and Methods - 2.6. RNA-seq analysis - is written in one sentence with a reference to a previous study. This is inappropriate. I recommend listing the essential points of the study so that the reader knows the basic parameters of the experiment (the details can already be found)! Results - not all Figures (e.g. 9) contain a self-explanatory legend, i.e. explanation of all abbreviations used. It is necessary to supplement. Discussion - is OK. References - it is necessary to carry out a careful check, especially in the names of journals, and to unite them according to the instructions for authors, and at the same time to respect the writing of capital and small letters in their names.

Reviewer 2 Report

Comments and Suggestions for Authors

 The article provides a comprehensive exploration of the MYB gene family in spinach and its role in various biological processes, including circadian rhythm, metabolism, and flower development. The following points highlight the strengths and comments on areas that need improvement:

Clarity and Organization:

The article is well-structured, with clear headings and a logical flow of information. This aids in understanding the research approach and findings.

Identification of MYB Genes:

The identification of 140 MYB genes in spinach is a significant achievement. It provides valuable insights into the diversity of MYB genes in this plant species, contributing to our understanding of gene regulation.

Genomic Localization:

The observation of MYB genes enriched at both ends of the sex chromosome (chromosome 4) is intriguing. It would be helpful to provide further explanations or hypotheses regarding the significance of this distribution.

Phylogenetic Analysis:

The phylogenetic analysis and comparison of 2R-MYB and 1R-MYB genes are informative. However, the article could benefit from more detailed discussion on why 2R-MYB genes exhibit higher conservation and what implications this has for plant biology.

Evolutionary Insights:

The discussion of tandem duplication and collinearity driving the evolution of spinach MYB genes is interesting. However, it would be helpful to discuss how these evolutionary mechanisms might relate to specific biological functions of MYB genes in spinach.

Subcellular Localization and Cis-acting Elements:

The prediction of subcellular localization and analysis of cis-acting elements provide functional insights. Elaboration on how these findings relate to the overall regulation of spinach growth and development would enhance the article.

Transcriptome Analysis:

The identification of candidate genes with biased expression in male spinach flowers is noteworthy. Further exploration of the potential roles of these genes in sex determination and anther development would be valuable.

Conclusion:

The conclusion summarizing the study's contributions to understanding MYB TFs and sex determination in spinach is appropriate. However, it could be expanded to emphasize the broader implications of the research.

In conclusion, this article makes a valuable contribution to the field of plant biology by characterizing MYB genes in spinach. Addressing the suggested points could enhance the overall impact and clarity of the research findings.

Reviewer 3 Report

Comments and Suggestions for Authors

Dear Authors,

I suggest modifying the title of the paper, since the analysis of MYB TF in the context of sex determination is more or less marginal (as is evident from the overall context of the manuscript).

The Latin botanical names of the species must include the abbreviation of the author. Latin names of plant families  (Amaranthaceae etc., lines 376, 377) must be in italics.

Within the Material and Methods is necessary to include the links of all applied tools (MUSCLE, MEGA7, TBtools).

Figures must include the information on the bioinformation tool applied.

In the discussion the authors confront the obtained results with 5 publications. Most of this part of the manuscript describes the results obtained. The discussion needs to be extended.

In the section 2.6 authors refere to the RNA-seq data originated from female and male flowers at five stages from study of Hulse-Kemp et al. (2021). This study has been done on monoecious spinach cv. Viroflay, where RNA originated from 17 tissues and different treatments. Please, declare the origine of RNA-seq data from female and male flowers. It is necessary to include the link of deposite database and accession numbers of BioProject and BioSample.

Round 2

Reviewer 1 Report

Comments and Suggestions for Authors

The authors accepted all my comments. Now I have no further comments and I recommend the manuscript for publication.

Reviewer 3 Report

Comments and Suggestions for Authors

Dear Authors,

thank you for considering and incorporating comments and suggestions for improving the quality of the manuscript.

Within the entire manuscript (including the title), I still recommend you to check the correct writing of the Latin botanical names of the species. It is also necessary to indicate the author's abbreviation, e.g. Spinacia oleraceae L.; Arabidopsis thaliana (L.) Heynh. etc..

It is also necessary to unify the form of listing species within one sentence, e.g. lines 57; 68-69; 182-183 etc., where you sometimes write Latin names and sometimes English names.

It is not necessary to repeatedly list a complete Latin name in the form of e.g. Arabidopsis thaliana, A. thaliana is sufficient. In the line 408 is missing dot (S. oleracea L.).

Line 472 Ricaceae family not Ricaceae family

Line 473 cruciferous family in not the family, it is a type of vegetables. The family, from the botanical meaning, always has the ending "eae".
